# Immunogenicity and Protective Capacity of Sugar ABC Transporter Substrate-Binding Protein against *Streptococcus suis* Serotype 2, 7 and 9 Infection in Mice

**DOI:** 10.3390/vaccines12050544

**Published:** 2024-05-15

**Authors:** Zujie Yan, Ruyi Pan, Junjie Zhang, Jianhe Sun, Xiaochun Ma, Nihua Dong, Xiaohui Yao, Jianchao Wei, Ke Liu, Yafeng Qiu, Katie Sealey, Hester Nichols, Michael A. Jarvis, Mathew Upton, Xiangdong Li, Zhiyong Ma, Juxiang Liu, Beibei Li

**Affiliations:** 1Shanghai Veterinary Research Institute, Chinese Academy of Agricultural Science, Shanghai 200241, China; yanzujie911@163.com (Z.Y.); panruyi614@163.com (R.P.); 17317271403@163.com (J.Z.); maxiaochun126@126.com (X.M.); dongnihua0519@163.com (N.D.); 15160870804@163.com (X.Y.); jianchaowei@shvri.ac.cn (J.W.); liuke@shvri.ac.cn (K.L.); yafengq@shvri.ac.cn (Y.Q.); zhiyongma@shvri.ac.cn (Z.M.); 2College of Veterinary Medicine, Hebei Agricultural University, Baoding 071000, China; 3Shanghai Key Laboratory of Veterinary Biotechnology, Shanghai Jiao Tong University, Shanghai 200240, China; sunjhe@sjtu.edu.cn; 4School of Cellular and Molecular Medicine, University of Bristol, University Walk, Bristol BS8 1TD, UK; katie.sealey@bristol.ac.uk; 5The Vaccine Group Ltd., Plymouth PL6 8BU, UK; hester.nichols@thevaccinegroup.co.uk (H.N.);; 6School of Biomedical Sciences, University of Plymouth, Plymouth PL4 8AA, UK; mathew.upton@plymouth.ac.uk (M.U.); 007352@yzu.edu.cn (X.L.); 7College of Veterinary Medicine, Yangzhou University, Yangzhou 225009, China

**Keywords:** *Streptococcus suis*, subunit vaccine, sugar ABC transporter substrate-binding protein, serotypes, universal vaccine

## Abstract

**Background:** *Streptococcus suis* (*S. suis*) is a Gram-positive bacterium that causes substantial disease in pigs. *S. suis* is also an emerging zoonoses in humans, primarily in Asia, through the consumption of undercooked pork and the handling of infected pig meat as well as carcasses. The complexity of *S. suis* epidemiology, characterized by the presence of multiple bacterial serotypes and strains with diverse sequence types, identifies a critical need for a universal vaccine with the ability to confer cross-protective immunity. Highly conserved immunogenic proteins are generally considered good candidate antigens for subunit universal vaccines. **Methods:** In this study, the cross-protection of the sugar ABC transporter substrate-binding protein (S-ABC), a surface-associated immunogenic protein of *S. suis*, was examined in mice for evaluation as a universal vaccine candidate. **Results:** S-ABC was shown to be highly conserved, with 97% amino acid sequence identity across 31 *S. suis* strains deposited in GenBank. Recombinantly expressed S-ABC (rS-ABC) was recognized via rabbit sera specific to *S. suis* serotype 2. The immunization of mice with rS-ABC induced antigen-specific antibody responses, as well as IFN-γ and IL-4, in multiple organs, including the lungs. rS-ABC immunization conferred high (87.5% and 100%) protection against challenges with *S. suis* serotypes 2 and 9, demonstrating high cross-protection against these serotypes. Protection, albeit lower (50%), was also observed in mice challenged with *S. suis* serotype 7. **Conclusions:** These data identify S-ABC as a promising antigenic target within a universal subunit vaccine against *S. suis*.

## 1. Introduction

*Streptococcus suis* (*S. suis*) is a pathogenic encapsulated Gram-positive bacterium that commonly resides within the upper respiratory tract of pigs. The bacterium can cause meningitis, arthritis, endocarditis, and other pathologies in a proportion of animals [1,2]. *S. suis* is also a zoonotic agent. Sporadic infection occurs in people through the consumption of undercooked pork and contact with *S. suis*-infected pigs or pork-derived products [3,4]. Currently, at least 29 different *S. suis* serotypes have been described based on the antigenicity of the bacterial capsular polysaccharides. These serotypes are distributed widely through pig herds, but not all serotypes or all strains within a serotype have the potential to cause disease. Worldwide, the predominant *S. suis* serotypes isolated from pigs include serotypes 1/2, 2, 3, 7, and 9 [5], with serotype 2 being the dominant serotype in pigs and most commonly associated with zoonotic disease [6].

Several strategies are used to control *S. suis* infection in pig herds. Antibiotic treatments are currently the mostly commonly used strategy; however, the emergence of drug-resistant strains and restrictions on the prophylactic use of antimicrobials in livestock production have increased interest in alternative strategies, primarily vaccination. Autogenous vaccines comprised of inactivated whole-bacteria (bacterin) *S. suis* are increasingly available for use in the field [6]; however, as these vaccines are ‘matched’ to *S suis* strains circulating in individual farms, they are serotype- or strain-specific and unable to provide cross-protection [6,7,8]. Multiple serotypes and divergent strains are present in pig herds [9], further complicating *S. suis* control and management by existing vaccines. Together, this complicated landscape has increased interest in multivalent or universal vaccines that can provide a stronger degree of cross-protection.

Host protection against *S. suis* infection is believed to be mediated primarily through opsonophagocytosis [6] by antibodies specific to bacterial surface components [10,11]. Commensurate with this mechanism of immunological control, surface-associated immunogenic proteins that are well conserved in multiple serotypes of *S. suis* are attractive targets for the development of universal subunit vaccines. In the past decade, a number of surface-associated immunogenic proteins of *S. suis* have been identified and evaluated as potential universal subunit vaccine candidates [6,12]. There are several proteins, such as muramidase-released protein (MRP), extracellular protein factors, parvulin-like peptidyl-prolyl isomerases (PrsA), phosphate ABC transporter ATP-binding protein (PstB), enolase, and Sao, have all been evaluated in mouse and pig models for the induction of protective immunity against homologous as well as heterologous strains of *S. suis* [13,14,15,16,17,18].

Sugar ABC transporter substrate-binding protein (S-ABC; alternatively called ABC transporter substrate-binding protein—maltose/maltodextrin or extracellular solute-binding protein) is a surface-associated immunogenic protein that was previously identified from the *S. suis* serotype 2 ZYS strain via immunoproteomic analysis [19]. In our current study, we examined the conservation and prevalence of S-ABC in different *S. suis* strains and evaluated the protective capacity of recombinant S-ABC (rS-ABC) against challenges of *S. suis* serotypes 2, 7, and 9 in mice. We show that the immunization of mice with rS-ABC induced antigen-specific antibody responses, as well as IFN-γ and IL-4 production, in multiple organs, and conferred protection against challenges with *S. suis* serotypes 2, 7, and 9.

## 2. Materials and Methods

### 2.1. Ethics Statement

All animal experiments were approved by the Institutional Animal Care and Use Committee of the Shanghai Veterinary Research Institute, China (IACUC No: SHVRI-SZ-2019070603), and performed in compliance with the Guidelines on the Humane Treatment of Laboratory Animals (Ministry of Science and Technology of the People’s Republic of China, Policy No. 2006 398).

### 2.2. Expression and Purification of Recombinant Proteins

Primers (5′-CGGAATTCATGAAACACAATCTCCTTAA-3′ and 5′-GCAAGCTTCTAGTTGCTGTGTTTTTGAG-3′) for cloning the S-ABC gene were designed based on the S-ABC sequence of the *S. suis* 05ZYH33 strain (GenBank: CP000407.1). The coding region of the S-ABC gene was amplified from the *S. suis* serotype 2 ZY05719 strain via a polymerase chain reaction (PCR) with the primers and inserted into a prokaryotic express vector, pET-28a, to generate a plasmid expressing His-tagged rS-ABC. A DNA construct encoding B cell epitopes of S-ABC linked by glycine (GGGG) linkers was chemically synthetized (Shanghai Personalbio Technology Co., Ltd., Shanghai, China) and inserted into the pET-28a vector to generate a plasmid expressing a His-tagged recombinant fusion construct comprised of B cell epitopes of S-ABC (rS-ABC-epitopes). All recombinant plasmids were confirmed via DNA sequencing. The rS-ABC and rS-ABC-epitope constructs were expressed in *Escherichia coli* BL21 (DE3) cells. Isopropyl-β-D-thiogalactopyranoside (IPTG) was used for induction, followed by Ni–NTA column purification according to the manufacturer’s protocol (SMART, Changzhou, China). Purified rS-ABC and rS-ABC-epitope proteins were then enriched using a 10 kDa cut-off Amicon Ultra Centrifugal (AUC) unit (EMD Millipore Billerica, MA, USA), followed by desalting with a 3 kDa cut-off unit. Centrifugation at 3000× *g* at 4 °C for 15 min was used in both cases. The resulting recombinant proteins were visualized by SDS-PAGE, quantified using an enhanced BCA protein assay kit (Beyotime biotechnology, Shanghai, China), and stored at −80 °C.

### 2.3. Multiple Sequence Alignment

Amino acid sequences of S-ABC from 31 *S. suis* strains were downloaded from GenBank. Multiple sequence alignment was performed using ClustalW software 2.1 (https://www.genome.jp/tools-bin/clustalw (accessed on 30 June 2021)) and visualized using ESPript software 3.0 (https://espript.ibcp.fr/ESPript/cgi-bin/ESPript.cgi (accessed on 30 June 2021)).

### 2.4. Prediction of Linear B Cell Epitopes

ABCpred (http://crdd.osdd.net/raghava/abcpred/ABC_submission.html (accessed on 21 July 2021)) and IEDB (http://tools.iedb.org/main/bcell/ (accessed on 21 July 2021)) bioinformatics software were used to predict linear epitopes within S-ABC. Based on these predictions, putative epitopes were analyzed for their immunogenicity and protective capacity.

### 2.5. Screening for Immunodominant Linear B Cell Epitopes with an Enzyme-Linked Immunosorbent Assay (ELISA)

Putative S-ABC B cell epitopes were chemically synthesized (Shanghai Apeptide Co., Ltd., Shanghai, China). Peptides were then screened using *S. suis*-specific sera with an ELISA. Briefly, peptides were coated onto ELISA plates at a concentration of 1 μg/mL, 5 μg/mL, and 10 μg/mL at 4 °C overnight. Plates were then blocked with 5% bovine serum albumin, followed by incubation with rabbit sera specific to *S. suis* serotype 2 (1:500 dilution) at 37 °C for 1 h. Antibodies bound to the peptides were detected via incubation with horseradish peroxidase-conjugated goat anti-rabbit IgG (Santa Cruz Biotechnology, Santa Cruz, CA, USA), and subsequently with 3,3′,5,5′-tetramethylbenzidine substrate. The optical density (OD_450nm_) of each well coated with the peptides was measured and normalized to the value of control wells. A relative OD_450nm_ value of >2.1 was considered positive.

### 2.6. Western Blot Analysis

Western blot for the detection of rS-ABC and rS-ABC-epitope proteins was performed with anti-His antibodies (Abcam, Shanghai, China) and rabbit sera specific to *S. suis* serotype 2, respectively, as described previously [20]. The rabbit sera specific to *S. suis* serotype 2 were obtained from convalescent rabbits experimentally infected with *S. suis* serotype 2.

### 2.7. Immunization and Protection Assay

Three-week-old BALB/c strain mice, purchased from Shanghai SLAC Laboratory Animal Co. Ltd. (Shanghai, China), were randomly divided into immunized and control groups (8 mice per group). Mice were vaccinated as indicated with 50 μg of purified rS-ABC or rS-ABC-epitope proteins emulsified in Freund’s complete adjuvant, administered subcutaneously. At 14 days, mice were boosted with 50 μg of an identical protein inoculum, but emulsified in Freund’s incomplete adjuvant. The control group was injected with phosphate-buffered saline (PBS) emulsified in Freund’s adjuvant. Depending on the study, groups were challenged intraperitoneally with an 8 × 50% lethal dose (LD_50_) of either the *S. suis* serotype 2 ZY05719 strain, the serotype 7 SH04815 strain, or the serotype 9 SH26 strain, at 10 days post-booster immunization and monitored for 7 days.

### 2.8. Detection of Antibodies with an ELISA

Blood samples were collected from mice at times indicated for the analysis of antibodies against the recombinant S-ABC proteins with an ELISA, as described previously [13]. Briefly, purified rS-ABC or rS-ABC-epitope proteins were coated onto ELISA plates at 1 μg per well and incubated at 4 °C overnight. The plates were blocked with 5% bovine serum albumin and then incubated with the sera collected from the immunized mice at 1:800 dilution. Antibodies bound to the recombinant proteins were detected via incubation with horseradish peroxidase-conjugated goat anti-mouse IgG (Santa Cruz Biotechnology) and subsequently with 3,3′,5,5′-tetramethylbenzidine substrate. The optical density (OD_450nm_) of each well was measured at 450 nm.

### 2.9. Detection of IL-4 and IFN-γ

At 10 days post-boost (24 days post-primary immunization), tissue samples (blood, lungs, and spleen) were collected. Levels of IL-4 and IFN-γ in tissues were quantified using corresponding ELISA kits (Boster, Wuhan, China) according to the manufacturer’s protocol.

### 2.10. Statistical Analysis

Measured values are presented as the means ± standard deviations (SDs). The significance of the results was determined by using Student’s *t*-tests. A value of *p* < 0.05 was considered statistically significant.

## 3. Results

### 3.1. Conservation and Prevalence of S-ABC across S. suis Serotypes

Although there is geographic variation, *S. suis* serotype 2 followed by 7 and 9 are predominant serotypes associated with disease in pig herds [6,9,21]. Amino acid sequences of S-ABC from 31 *S. suis* strains, including 24 serotype 2, 2 serotype 7, and 5 serotype 9, available in GenBank were downloaded and aligned, followed by a conservation analysis. Multiple sequence alignments detected a total of 15 amino acid variations across the entire 419-amino-acid-residue S-ABC protein, representing 97% sequence identity among the *S. suis* strains (Appendix A). These data suggest that S-ABC is highly conserved across clinically important *S. suis* serotypes and strains.

### 3.2. Immunodominant Linear B Cell Epitopes of S-ABC

S-ABC was previously identified via an immunoproteomic analysis with swine sera specific to *S. suis* serotype 2, indicating expression and exposure of the protein to the immune system during the natural infection of pigs [19]. The 419-amino-acid S-ABC protein is primarily a hydrophilic protein without a transmembrane domain. Results from a predictive antigenicity analysis using Protean software version 7.1 and VaxiJen version 2.0 ere consistent with the observed immunogenicity of the S-ABC protein (Appendix A) [22]. The S-ABC sequence was then analyzed using ABCpred and IEDB bioinformatic tools to predict potential linear B cell epitopes. Results from this analysis, which identified five putative B cell epitopes, are shown in Figure 1A and Table 1.

As with PtsB [13], peptides representing the five predicted linear B cell epitopes of S-ABC were chemically synthesized in order to identify the immunodominant linear B cell epitopes. The peptides were coated onto ELISA plates and reactivity was analyzed using hyperimmune rabbit sera specific to *S. suis* serotype 2. Of the five predicted B cell epitopes, peptide EP1 (SLDNQSGSAPDVMMAPY) and EP5 (ANTEAREYAVSKKDEL) were shown to be recognized by the rabbit sera (Table 1 and Figure 1B). Based on this result, EP1 and EP5 were considered immunodominant, and used for the analysis of immunogenicity and protective capacity in mice.

### 3.3. Preparation of rS-ABC and rS-ABC-Epitope Proteins

B cell epitope EP1 and EP5 of S-ABC were tandemly connected by a linker of GGGG (Figure 2A) and expressed as a multiple-epitope S-ABC construct (called rS-ABC-epitope). His-tagged rS-ABC (full-length protein) and rS-ABC-epitope were expressed in *E. coli* and purified by using a Ni–NTA column (Figure 2B). Purified rS-ABC and rS-ABC-epitope expression was confirmed by a Western blot analysis using an anti-His antibody (Figure 2C, Anti-His panel). To examine their ability to be recognized by *S. suis*-specific sera, the purified rS-ABC and rS-ABC-epitope were examined by a Western blot analysis using the rabbit sera specific to *S. suis* serotype 2. Although both recombinant proteins were recognized by the rabbit sera, antibody binding to rS-ABC was at a much higher level compared to rS-ABC-epitope, even though a comparison of the anti-His reactivity would indicate a greater relative concentration of the rS-ABC-epitope protein (Figure 2C). Together with earlier work [19], these results indicate that S-ABC induces an antibody response during *S. suis* infection, which appears to be partially localized to two immunodominant B cell epitopes within the protein, and identifies S-ABC as a potential target for development as a subunit vaccine.

### 3.4. Levels of Antibodies and Cytokines in Immunized Mice

To assess the immunogenicity of the two S-ABC constructs, mice were immunized with either rS-ABC or rS-ABC-epitope protein as outlined in the schematic (Figure 3A). The corresponding anti-rS-ABC or rS-ABC-epitope antibody response was then analyzed at times indicated by an ELISA using rS-ABC or rS-ABC-epitope protein as the antigen. As shown in Figure 3B, high levels of antibodies specific to rS-ABC were detected in rS-ABC-vaccinated mice at 14 days post-primary vaccination, which increased after the boost. In contrast, no antibodies to rS-ABC-epitope were observed in the rS-ABC-epitope vaccinated mice at 14 days post-primary vaccination; rS-ABC-epitope-specific antibodies then increased to only a relatively low level following the day 14 boost (Figure 3B). Interestingly, this result was consistent with the difference in antigenicity observed for the two S-ABC constructs using the *S. suis* serotype 2-specific rabbit sera (Figure 2C), suggesting differences in both immunogenicity as well as antigenicity.

Th1-type and Th2-type immune responses play an important role in the control of *S. suis* infection [6]. IFN-γ and IL-4 are critical cytokines involved in the induction of Th1 and Th2 responses, respectively [23,24], and have been used as indicators for these responses in animals in *S. suis* vaccination studies [15,23,25,26,27]. We therefore examined the induction of IFN-γ and IL-4 in the immunized mice. Sera, lung, and spleen tissues were collected from immunized mice at 10 days post-boost and analyzed for IFN-γ and IL-4 levels with an ELISA. The immunization of mice with rS-ABC or rS-ABC-epitope proteins induced a significantly higher level of both IFN-γ and IL-4 production in the sera, as compared with the PBS control mice (Figure 3C). IFN-γ and IL-4 levels in the different groups paralleled the antibody response seen in Figure 3B, with both IFN-γ and IL-4 being substantially higher in mice immunized with rS-ABC compared to mice receiving rS-ABC-epitope (Figure 3C). Similar differences in IFN-γ and IL-4 induction by the two constructs were observed in the lungs (Figure 3D) and spleen (Figure 3E). Overall, both rS-ABC and rS-ABC-epitope were able to induce antigen-specific antibodies as well as IFN-γ and IL-4 production at multiple tissue sites, but rS-ABC was by far the more immunogenic in all measures of immune response induction.

### 3.5. Protective Efficacy of rS-ABC and rS-ABC-Epitope

The next series of experiments were designed to determine whether the observed differences in immunogenicity translated to differences in protection using both homologous as well as divergent, heterologous *S. suis* serotypes. Mice were immunized with rS-ABC or rS-ABC-epitope and challenged with a lethal dose (8 × LD_50_) of *S. suis* serotype 2, 7, or 9 (Figure 3A). The survival analysis presented in Figure 4 showed that rS-ABC provided substantial levels of protection (87.5% and 100%) against *S. suis* serotypes 2 and 9, with protection dropping to 50% for *S. suis* serotype 7. Consistent with the immunogenicity results, survival in rS-ABC-epitope-vaccinated mice was consistently lower, with levels of protection against the different serotypes ranging from 25% to 37.5% (Figure 4). Overall, these data indicate that rS-ABC was able to provide high levels of protection against *S. suis* serotype 2 and 9 challenges, and lower levels against *S. suis* serotype 7 challenges, with the full-length rS-ABC protein being far superior at all levels to the epitope-focused rS-ABC-epitope construct.

## 4. Discussion

Subunit vaccines based on well-conserved bacterial surface-associated immunogenic proteins are considered potential universal vaccine candidates with which to confer cross-protection against different *S. suis* serotypes [5]. In this study, we examined the antigenicity/immunogenicity and protective capacity of S-ABC against *S. suis* serotypes 2, 7, and 9 in mice.

S-ABC is an immunogenic cell-wall-associated protein that was identified as a potential vaccine candidate by the immunoproteomic analysis of *S. suis* serotype 2 (ZYS strain) [19]. S-ABC is recognized by both swine sera generated against formaldehyde-inactivated *S. suis* ZYS strains as well as by convalescent sera from pigs infected in the field by *S. suis* serotype 2, suggesting that this protein is a potential target for the development of *S. suis* subunit vaccines [19]. To evaluate this potential, we analyzed the level of amino acid sequence conservation and prevalence of S-ABC across different *S. suis* strains. Multiple sequence alignment indicated that S-ABC was highly conserved with 97% amino acid sequence identity among 31 *S. suis* strains, including 24 serotype 2, 2 serotype 7, and 5 serotype 9, which are the predominant serotypes in pig herds [6,9,21]. The high conservation combined with the earlier serological data from pigs supported its potential use as a universal subunit vaccine candidate [19].

To examine this possibility, the S-ABC gene was cloned from the *S. suis* serotype 2 ZY05719 strain and expressed in *E. coli*. The rS-ABC was recognized by *S. suis* serotype 2-specific sera from convalescent rabbits experimentally infected with *S. suis* serotype 2, indicating that S-ABC was an immunogenic protein during *S. suis* infection, in agreement with the previous observation from pigs [19]. The immunization of mice with rS-ABC induced high levels of antibody production, which was detectable in the rS-ABC-immunized mice at 14 days post-primary vaccination, and increased after a boost, suggesting that rS-ABC was able to elicit an antigen-specific antibody response in animals. Th1- and Th2-type immunity, and their associated cytokines (IFN-γ and IL-4, respectively), have been shown to contribute to the resistance against *S. suis* infection [15,23,25,26,27]. Immunized mice with rS-ABC induced high levels of IFN-γ and IL-4 production, suggesting that rS-ABC was able to induce Th1- and Th2-type immune responses that are essential for the restriction of *S. suis* infection.

To ascertain the functional relevance of the S-ABC immunity induced by vaccination, we challenged the immunized mice with *S. suis* serotypes 2, 7, and 9 to evaluate the protective efficacy of rS-ABC. *S. suis* serotype 2 is the most virulent and prevalent serotype in pig and human infection worldwide [6], while *S. suis* serotypes 7 and 9 are predominant serotypes in pig infection [21]. A survival analysis showed that rS-ABC conferred the immunized mice 87.5% and 100% protection against challenges of *S. suis* serotype 2 and 9, respectively, suggesting a high cross-protection against *S. suis* serotype 2 and 9 infection. Lower protection was observed in the mice challenged with *S. suis* serotype 7, with a protection percentage of 50%. This lower protection against *S. suis* serotype 7 was probably attributable to the relatively higher virulence of the challenged strain, compared to the challenge strains of *S. suis* serotypes 2 and 9. Although the challenge doses (8 × LD_50_) were the same for all strains, the control mice challenged with *S. suis* serotype 7 died quickly (all control mice died within 36 h post-challenge) as compared to those challenged with *S. suis* serotypes 2 and 9 (all control mice died within 48 h post-challenge).

Bacterial surface-associated immunogenic proteins are attractive targets for the development of a subunit vaccine against *S. suis* infection [6,12]. Most of these proteins are important for *S. suis* biological processes, such as nutrient acquisition, metabolism, adhesion, invasion, and virulence [28]. More than 17 surface-associated immunogenic proteins of *S. suis* have been identified as potential candidates for the development of subunit vaccines [7,15]. As reviewed in our previous publication [13], proteins such as HtpS and SsPepO, highly conserved in *S. suis* serotype 2, were shown to be widely distributed and conferred protection against homologous *S. suis* serotype 2 challenges [29,30]; however, not all surface-associated immunogenic proteins are able to provide high protection against *S. suis* infection, with Lmb providing lower levels (50%) of protection even against *S. suis* serotype 2 homologous challenge [31]. Similarly to S-ABC, the enolase protein showed 100%, 80%, and 100% protection against *S. suis* serotypes 2, 7, and 9, respectively, in mice models [15]; however, even with surface-associated immunogenic proteins known to provide significant protection against *S. suis* infection, their cross-protective capacity against different *S. suis* serotypes is rarely investigated. These recent observations, together with our current data, suggest that only a subset of conserved surface-associated immunogenic proteins is suitable for the development of universal subunit vaccines capable of inducing responses against different *S. suis* serotypes.

It is known that several factors, such as the method of inactivation [8,32], number of vaccine doses [32], and adjuvant used in a formulation [8,13] are important for the protective efficacy of an *S. suis* vaccine [7]. For example, the enolase protein discussed above is described to be either protective or nonprotective depending on the particular study [33,34]. This difference in protective effect may be attributable to the different adjuvants used in the vaccine formulation; therefore, there may be several ways to improve the cross-protection efficacy of *S. suis* vaccines. Recently, a live attenuated *Salmonella enterica* serovar Choleraesuis vector was used to deliver the enolase, resulting in high cross-protection against *S. suis* serotypes 2, 7, and 9 in mice [15]. This observation suggests a novel way to improve the cross-protection efficacy of universal subunit vaccines against the multiple serotypes of *S. suis* infection by altering the antigen delivery system.

B cell epitopes are groups of residues on the surface of an antigen that stimulate humoral immune responses. Recent advances in bioinformatics have enabled the design of epitope-based vaccines against the infection of different microbials [35], such as classical swine fever virus [36], *Staphylococcus aureus*, and the Leishmania parasite [37,38]. In the case of *S. suis*, a B cell epitope-based subunit vaccine that consisted of a linear array of three epitopes of GAPDH, eight epitopes of MRP, and four epitopes of DLDH showed 90% protection against the challenge of the *S. suis* serotype 2 HA9801 strain in mice [39]. In the present study, we evaluated the protection capacity of a similar rS-ABC-epitope construct against challenges with *S. suis* serotype s2, 7, and 9. We observed 25%–37.5% protection against these serotypes, which was remarkably lower than the protection (50%-100%) induced by an intact full-length rS-ABC protein. The level of antigen-specific antibodies and cytokines (IFN-γ and IL-4) induced by rS-ABC-epitopes were also lower than those induced by the full-length version. These data indicate that the intact rS-ABC protein is superior to the multi-epitope rS-ABC-epitope in both the induction of immune responses and protection against *S. suis* infection. The rS-ABC-epitope was constructed using only the two major immunodominant linear B-cell epitopes of S-ABC. Intact proteins may contain multiple B cell epitopes, including those that are conformational. Additionally, CD4+ T cell epitopes critical for helper function during antibody induction and production are also necessary for robust immunity. The low protection of rS-ABC-epitope may be attributable to the low number of epitopes used and/or a lack of key immunodominant epitopes that were missed in this study. Overall, our data suggest the importance of considering these aspects during antigen design for subunit vaccines based on linear epitope arrays.

## 5. Conclusions

This study continues our focus on the identification of protein antigen candidates for inclusion in a universal *S. suis* vaccine [13]. Similar to other identified candidates, S-ABC was highly conserved across *S. suis* serotypes and strains, with 97% amino acid sequence similarity among the 31 *S. suis* strains examined. rS-ABC was recognized via the rabbit sera specific to *S. suis* serotype 2, and was able to elicit antigen-specific antibody responses as well as induce IFN-γ and IL-4 production in multiple tissues. Recombinantly expressed rS-ABC conferred 87.5% and 100% protection against challenges with *S. suis* serotypes 2 and 9, respectively. This supports S-ABC being a protein candidate with the ability to confer high levels of cross-protection against multiple *S. suis* serotypes, in this case *S. suis* serotypes 2 and 9. Medium protection (50%) was observed in the rS-ABC mice in the more aggressive *S. suis* serotype 7. Interestingly, the full-length S-ABC protein was associated with greater immunogenicity and protection than a construct comprised of a linear B cell epitope array, which is the same observed for PstB but the opposite for a GAPDH-MRP-DLDH epitope construct [13,39]. This suggests that the specific nature of the antigen may make it more or less amenable to use in synthetic poly-epitope antigen arrays. Together, these data suggest that S-ABC is an additional promising target for the development of a universal subunit vaccine against different *S. suis* serotype infections.

## Figures and Tables

**Figure 1 vaccines-12-00544-f001:**
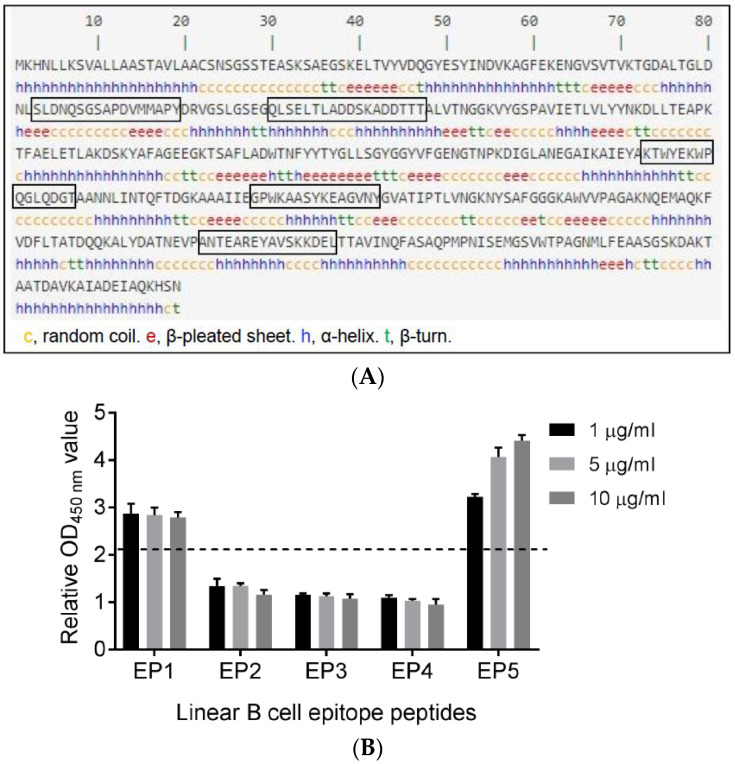
Predicted linear B cell epitopes of S-ABC. (**A**) Linear B cell epitopes of S-ABC were predicted using the bioinformatics software ABCpred and IEDB. The predicted B cell epitope sequences are boxed. (**B**) Screening for immunodominant linear B cell epitopes. The predicted B cell epitope peptides (EP1-5) were coated onto ELISA plates and incubated with rabbit sera specific to *S. suis* serotype 2. The OD_450nm_ value of each well coated with peptide was normalized to the value of control wells and plotted. A relative OD_450nm_ value of >2.1 was considered positive.

**Figure 2 vaccines-12-00544-f002:**
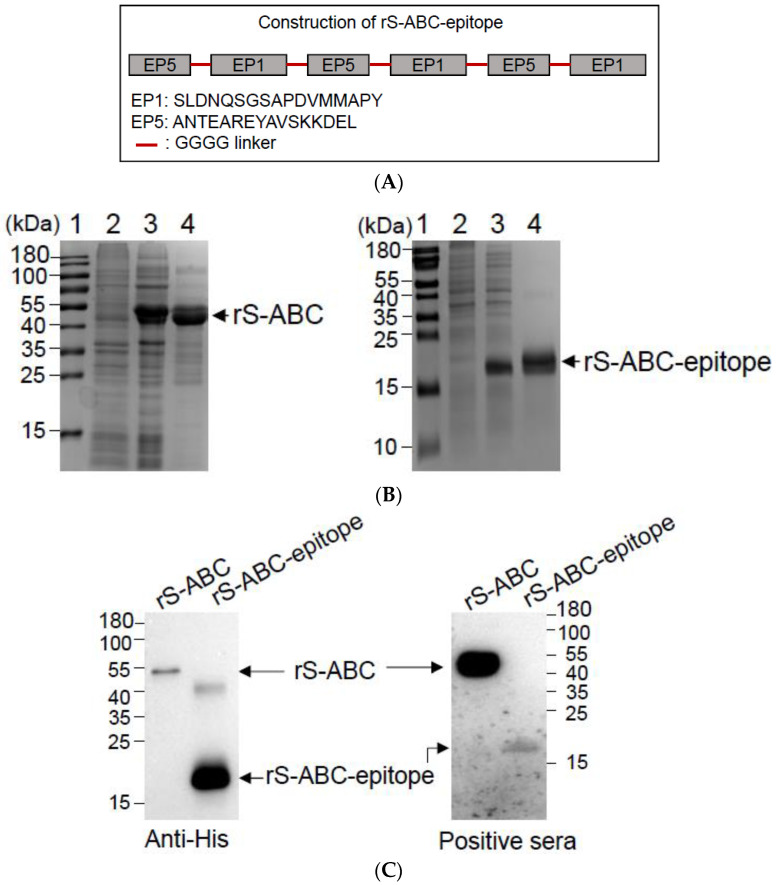
Construction and expression of rS-ABC and rS-ABC-epitope proteins. (**A**) Schematic representation of the construction of rS-ABC-epitope. EP1 and EP5 were tandemly connected by GGGG linkers to be expressed as a multiple-epitope protein. (**B**) Expression and purification of rS-ABC and rS-ABC-epitope proteins. The rS-ABC and rS-ABC-epitope proteins were expressed from *E. coli*, induced with IPTG, and purified via their His-tag by using a Ni–NTA column. The resulting recombinant proteins were then examined using SDS-PAGE. Lanes: 1, protein marker; 2, uninduced bacterial cells; 3, bacterial cells induced with IPTG; and 4, purified recombinant protein. (**C**) Antigenic analysis of rS-ABC and rS-ABC-epitope proteins. The rS-ABC and rS-ABC-epitope purified proteins were examined by a Western blot using anti-His antibodies and positive rabbit sera containing antibodies specific to *S. suis* serotype 2.

**Figure 3 vaccines-12-00544-f003:**
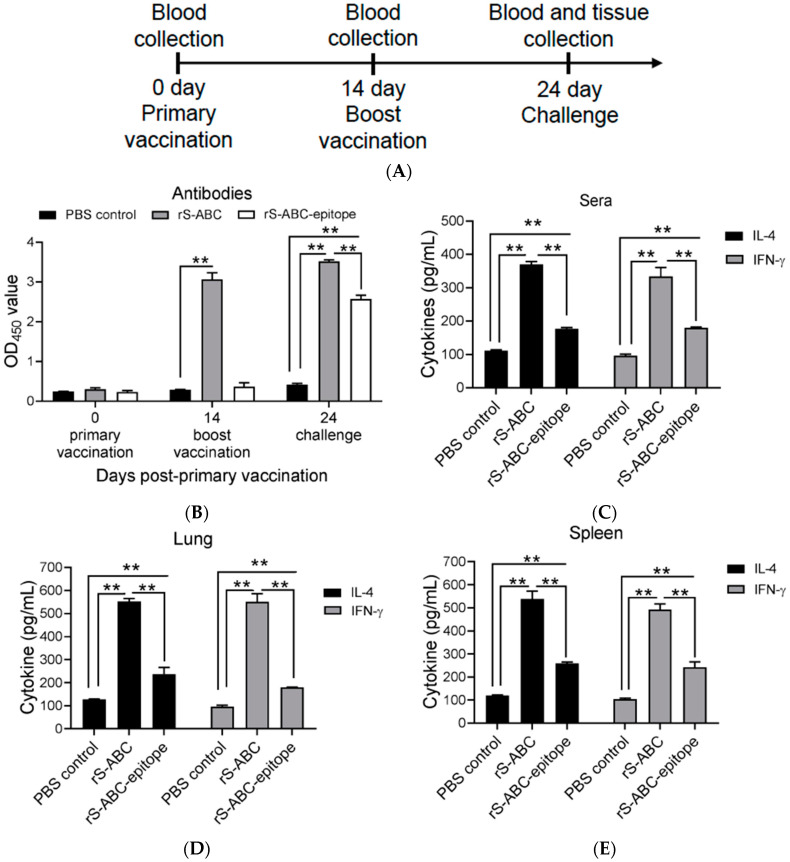
Levels of antibodies and cytokines in immunized mice. (**A**) Schematic representation of vaccination and sample collection. Mice were vaccinated with rS-ABC or rS-ABC-epitope proteins and then boosted at 14 days post-primary vaccination. (**B**) Levels of antibodies specific to rS-ABC or rS-ABC-epitope were examined with an ELISA in sera collected prior to primary vaccination, boost vaccination, and challenge. (**C**–**E**) Levels of IL-4 and IFN-γ in sera (B), lungs (**C**), and spleen (**D**) collected at 24 days post-primary vaccination were examined with an ELISA. Statistically significant differences between groups are indicated by asterisks (** *p* < 0.01).

**Figure 4 vaccines-12-00544-f004:**
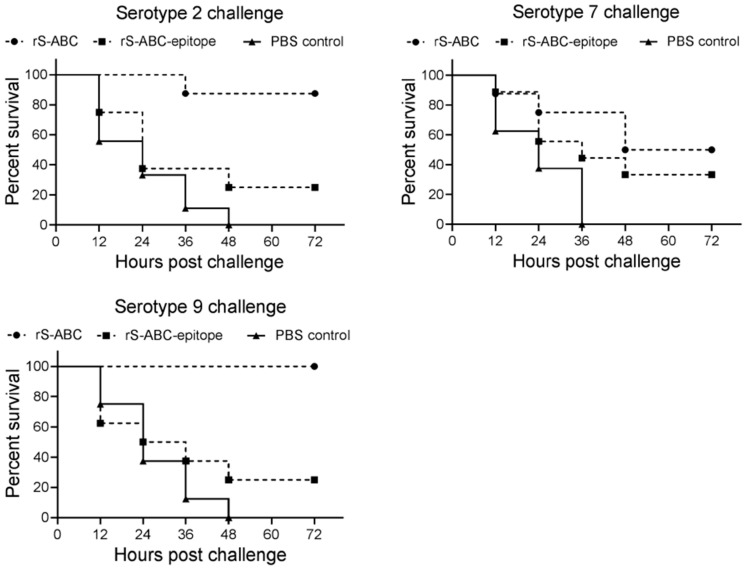
Protective efficacy of rS-ABC and rS-ABC-epitope in mice. Three-week-old BALB/c strain mice (8 mice per group) were vaccinated with rS-ABC or rS-ABC-epitope proteins and boosted at 14 days post-primary vaccination. Mice were challenged with a lethal dose of *S. suis* serotype 2, 7, or 9 at 10 days post-boost vaccination and monitored for 7 days. Animals succumbed to infection between 12 and 48 h post-challenge, and the survival curves were plotted.

**Table 1 vaccines-12-00544-t001:** Predicted linear B cell epitopes of S-ABC.

No.	Position	Amino Acid Sequence	Length	Recognized by Positive Sera
EP1	83-99	SLDNQSGSAPDVMMAPY	17	Yes
EP2	110-126	QLSELTADDSKADDTTT	17	No
EP3	233-247	KTWYEKWPQGLQDGT	15	No
EP4	268-282	GPWKAASYKEAGVNY	15	No
EP5	342-357	ANTEAREYAVSKKDEL	16	Yes

## Data Availability

The data can be shared up on request.

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
