# Peer review of "Immunogenicity and Protective Capacity of Sugar ABC Transporter Substrate-Binding Protein against Streptococcus suis Serotype 2, 7 and 9 Infection in Mice"

_vaccines, 2024, doi:10.3390/vaccines12050544_

Round 1

Reviewer 1 Report

Comments and Suggestions for Authors

Zujie Yan and cols. study in this manuscript the antigenic, immunogenic and protective capacities of a surface protein of the zoonotic bacterium Streptococcus suis in a mouse model. They perform first a multiple sequence alignment to demonstrate the conservation of the protein sequence and, after obtaining the recombinant protein, they carry out the antigenicity studies (in silico) as well as the immunogenicity and protection assays (in vitro and in vivo). The work is well performed and the manuscript is well written. However, although they explain that the protein had shown in a previous immunoproteomics study its immunoreactive nature, they do not justify properly why they have chosen this protein for their study, as there are also (many) other surface proteins with at least the same potential. It would be desirable to write the rationale of the work.

Other comments:
Introduction:
-Lines 68-71: there have been more protein candidates tested so far apart from those cited.

Results:
Section 3.1: What do the authors mean with the statement that the S-ABC gene is prevalent? The gene is known to be present in all the genomes sequenced so far. In my opinion, Figure 1B is not necessary.
The authors provide a supplementary Figure 1, but panels A (hydrophobicity analysis) and B (transmembrane domain analysis) are not commented at all in the main manuscript body. Please describe those results or remove them if you consider they are not necessary. Concerning transmembrane analysis, TMHMM does not detect TMH, as the protein is a lipoprotein (it contains the LAAC sequence that is characteristic of lipoproteins, as well as a signal peptide II).
Section 3.2: There are predictive antigenicity algorithms better than Protean. Please compare this with others that are specially for vaccine  design, like Vaxign or Jenner Predict.

Discussion:
-Line 357: I do not agree with the statement that enolase is a surface protein. This is a cytoplasmic protein (from the glycolysis pathway) that has been found on the surface of different bacterial species (therefore considered as a “moonlighting protein”), conferring protective capacity (depending on the study, as appointed by the authors).

Minor spelling typo errors:
“western blot” should be “Western blot” (W in capitals). Please revise it throughout the text.

Comments on the Quality of English Language

Well written in general. Minor grammar/syntax/typo errors.

Author Response

Review 1

Zujie Yan and cols. study in this manuscript the antigenic, immunogenic and protective capacities of a surface protein of the zoonotic bacterium Streptococcus suis in a mouse model. They perform first a multiple sequence alignment to demonstrate the conservation of the protein sequence and, after obtaining the recombinant protein, they carry out the antigenicity studies (in silico) as well as the immunogenicity and protection assays (in vitro and in vivo). The work is well performed and the manuscript is well written. However, although they explain that the protein had shown in a previous immunoproteomics study its immunoreactive nature, they do not justify properly why they have chosen this protein for their study, as there are also (many) other surface proteins with at least the same potential. It would be desirable to write the rationale of the work.
Response: Thanks so much for your comments and recognition of our work. The origin of this work indeed comes from that paper (Zhang A, Xie C, Chen H, et al. Proteomics, 2008, 8(17): 3506-3515). S-ABC was one of the cell wall-associated proteins identified by immunoproteomic analysis in that paper. Actually, the S-ABC was not the only one protein we selected for further test. We choose several potential proteins (not only from the Proteomics paper but also other papers and our own bioinformatic analysis) to test their antigenic, immunogenic and protective capacities. Some of them are usable, some not, and there are still some candidates are in the testing process. In the present paper, we only exhibited the data of one of the “useful” protein S-ABC. In the future, we hope find several protective proteins and develop a combined subunit vaccine for Streptococcus suis.  

Other comments:
Introduction:
-Lines 68-71: there have been more protein candidates tested so far apart from those cited.
Response: Thanks so much for your comments. We have added additional tested proteins in the sentence and also relevant references. Please see lines 67-71 and lines 432-438 in the revised manuscript.
Results:
Section 3.1: What do the authors mean with the statement that the S-ABC gene is prevalent? The gene is known to be present in all the genomes sequenced so far. In my opinion, Figure 1B is not necessary.

Response: Thanks so much for your questions. “prevalent” means the S-ABC gene is present in different S. suis strains, which means it could have the potential as universal subunit vaccine to provide protection to all S. suis strains. As your suggestion, the presence of the S-ABC gene in different S. suis strains could be demonstrated by the WGS data. We deleted the Figure 1B, the section 2.4 of the “Materials and methods” (PCR detection of the S-ABC gene) and relevant description in the revised manuscript. Moreover, based on reviewer 2’s comments, we put the figure 1(Multiple amino acid sequence alignment of S-ABC from S. suis serotype 2, 7 and 9) into the supplementary materials. Please see Lines 26-28, 111-112, 172-173, 293-297 and supplementary Figure 1.   

The authors provide a supplementary Figure 1, but panels A (hydrophobicity analysis) and B (transmembrane domain analysis) are not commented at all in the main manuscript body. Please describe those results or remove them if you consider they are not necessary. Concerning transmembrane analysis, TMHMM does not detect TMH, as the protein is a lipoprotein (it contains the LAAC sequence that is characteristic of lipoproteins, as well as a signal peptide II).

Response: Thanks a lot for your suggestion. We have deleted the panels A and B. Please see the modified supplementary Figure 2A.

Section 3.2: There are predictive antigenicity algorithms better than Protean. Please compare this with others that are specially for vaccine  design, like Vaxign or Jenner Predict.
Response: Thanks a lot for your comments. As your suggestion, we tried to analyze the antigenicity of the S-ABC protein with the Vaxign and Jenner Predict. However, both servers are not working. We can open the Vaxign 1.0 website (https://violinet.org/vaxign/), however, after we submit the protein sequence, it cannot return the result. For the Vaxign 2.0 and Jenner Predict, we cannot open the their websites (https://violinet.org/vaxign2 and http://117.211.115.67/vaccine/home.html). We tried several times, they did not work for us. Instead, we used another software program VaxiJen 2.0, a widely used server for prediction of bacterial protective antigens (Doytchinova, I. A., & Flower, D. R. (2007). BMC bioinformatics, 8, 1-7. Dalsass, M., Brozzi, A., Medini, D., & Rappuoli, R. (2019). Frontiers in immunology, 10, 431583.). The VaxiJen 2.0 give a score 0.4862, which means it is a Probable ANTIGEN. Please see supplementary Figure 2B.  

Discussion:
-Line 357: I do not agree with the statement that enolase is a surface protein. This is a cytoplasmic protein (from the glycolysis pathway) that has been found on the surface of different bacterial species (therefore considered as a “moonlighting protein”), conferring protective capacity (depending on the study, as appointed by the authors).
Response: Thanks a lot for providing the information of the enolase. We have deleted the description “a conserved surface protein of S. suis”. Please see lines 340-341 in the revised manuscript.

Minor spelling typo errors:
“western blot” should be “Western blot” (W in capitals). Please revise it throughout the text.

Response: Thanks so much for your reminding. We have modified this throughout the text in the revised manuscript.

Reviewer 2 Report

Comments and Suggestions for Authors

The manuscript entitled “Immunogenicity and protective capacity of sugar ABC transporter substrate-binding protein against Streptococcus suis serotype 2, 7 and 9 infection in mice” is well written and well structured by the Authors who highlight considerable scientific rigor in every section of the research.

The experimental plan is well structured and described in each step in a very clear and exhaustive way.

However, some changes should definitely be considered to improve reader understanding.

First of all, part "a" of figure 1 is absolutely illegible. The Authors should find a way to make it clearly visible for correct interpretation by the reader, otherwise think of inserting it only as supplementary material to be referred to for reference.

The file uploaded with the original images, which are also very interesting, should be accompanied by an explanatory legend.

I would suggest that the Authors keep in mind for discussion the important acquisitions of other researchers such as

Dresen, Muriel, Peter Valentin-Weigand, and Yenehiwot Berhanu Weldearegay. "Role of metabolic adaptation of Streptococcus suis to host niches in bacterial fitness and virulence." Pathogens 12.4 (2023): 541.

Author Response

Review 2

The manuscript entitled “Immunogenicity and protective capacity of sugar ABC transporter substrate-binding protein against Streptococcus suis serotype 2, 7 and 9 infection in mice” is well written and well structured by the Authors who highlight considerable scientific rigor in every section of the research.

The experimental plan is well structured and described in each step in a very clear and exhaustive way.

However, some changes should definitely be considered to improve reader understanding.

First of all, part "a" of figure 1 is absolutely illegible. The Authors should find a way to make it clearly visible for correct interpretation by the reader, otherwise think of inserting it only as supplementary material to be referred to for reference.

Response: Thanks a lot for your comments. As your suggestion, we put the Figure 1a into supplementary material as supplementary Figure 1. We will provide high-revolution pictures to make it clear for reader.

The file uploaded with the original images, which are also very interesting, should be accompanied by an explanatory legend.

Response: Thanks a lot for your comment. We have modified the original images as your comments. Since the figures’ name and order in the revised manuscript have been changed based on the reviewer’s comments, we have re-uploaded the original pictures as a single PDF file, in which the corresponding to the figures in the main article, labels of the lanes and explanatory legend was exhibited.  

I would suggest that the Authors keep in mind for discussion the important acquisitions of other researchers such as

Dresen, Muriel, Peter Valentin-Weigand, and Yenehiwot Berhanu Weldearegay. "Role of metabolic adaptation of Streptococcus suis to host niches in bacterial fitness and virulence." Pathogens 12.4 (2023): 541.

Response: Thanks a lot for your suggestion. We have added some description in the “discussion” section of the revised manuscript based on your recommended paper and also added this paper in the references. Please see lines 326-328 and 458-459.

Round 2

Reviewer 1 Report

Comments and Suggestions for Authors

The authors have adequately addressed the comments and suggestions made by the reviewers.

Author Response

Thanks a lot for your comments. Based on the comments from editor, We have made the following modification for the manuscript. 

(1) We have carefully checked all references and confirmed that they are all correct in the revised manuscript. All the revised reference numbers were marked with blue color.

(2) The similarity of the highlighted text shown in the PDF file was carefully modified and reduced. Please see the revised manuscript marked with yellow color.